# Automated Classification Analysis of Geological Structures Based on Images Data and Deep Learning Model

**Ye Zhang [1], Gang Wang [2], Mingchao Li [1,\*] and Shuai Han [1]**

[1]   State Key Laboratory of Hydraulic Engineering Simulation and Safety, Tianjin University, Tianjin 300354, China; jgzhangye@tju.edu.cn (Y.Z.); hs2015205039@tju.edu.cn (S.H.)
[2]   Chengdu Engineering Corporation Limited, PowerChina, Chengdu 610072, China; wgcd126a@126.com
\*   Correspondence: lmc@tju.edu.cn

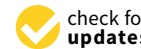

**Featured Application: This work aims to build a robust model with a comparison of machine learning, convolutional neural network and transfer learning. The model can be combined with an unmanned aerial vehicle (UAV) to act as a tool in geological surveys in the future.**

**Abstract:** It is meaningful to study the geological structures exposed on the Earth's surface, which is paramount to engineering design and construction. In this research, we used 2206 images with 12 labels to identify geological structures based on the Inception-v3 model. Grayscale and color images were adopted in the model. A convolutional neural network (CNN) model was also built in this research. Meanwhile, K nearest neighbors (KNN), artificial neural network (ANN) and extreme gradient boosting (XGBoost) were applied in geological structures classification based on features extracted by the Open Source Computer Vision Library (OpenCV). Finally, the performances of the five methods were compared and the results indicated that KNN, ANN, and XGBoost had a poor performance, with the accuracy of less than 40.0%. CNN was overfitting. The model trained using transfer learning had a significant effect on a small dataset of geological structure images; and the top-1 and top-3 accuracy of the model reached 83.3% and 90.0%, respectively. This shows that texture is the key feature in this research. Transfer learning based on a deep learning model can extract features of small geological structure data effectively, and it is robust in geological structure image classification.

**Keywords:** OpenCV; machine learning; transfer learning; Inception-v3; geological structure images; convolutional neural networks

## 1. Introduction

The primary objective of a geological survey is to identify geological structures in the field and, this is also important for project schedule management and safety guarantees. In construction, engineers search for the exposure of geological structures to the Earth's surface in field surveys, then explore geological structures that partly extend below the Earth's surface with boreholes, adits, etc. Some geological structures should be given special attention because of their poor properties. Anticline and ptygmatic folds weather easily; xenoliths, boudins, and dikes usually have low strength at the contact surface because they contain rocks with different properties; ripple marks, mudcracks, and concretion always indicate there is an ancient river course; faults and scratches mean broken structures in engineering; basalt columns have a low strength because of the columnar joints; a gneissose structure also has a low shear strength at the direction of schistosity. The geological structures have a significant influence on project site selection, general layout and schedule management, which is

also crucial to construction quality. The identification of geological structures can help engineers make a better choice in construction. On the other hand, geological structures, such as faults [1] and folds [2], are connected to hazards. Vasu and Lee [3] applied an extreme learning machine to build the landslide susceptibility model with 13 selected features (including geological structure features). The performance of prediction was better, with the accuracy of 89.45%. Dickson and Perry [4] explored three machine learning methods, namely maximum entropy models, classification and regression trees and boosted regression trees, to make identification of coastal cliff landslide control based on geological structure parameters. The final result showed a high performance with 85% accuracy. The researches proved geological structures were connected to geologic hazard prediction and prevention. However, the machine learning methods are applied in structured data. For unstructured data, such as image, audio, text, we need to extract the features of the unstructured data and input them to train machine learning models. In the process of data type transformation, other algorithms are going to be selected.

3D visualization techniques and data interpolation are also used in geological structure detection. Zhong et al. [5] made a 3D model of the complex geological structure based on borehole data. Discrete fractures in rocks were also computed and estimated by 3D model construction [6,7]. Spatial relationships of geological data were easy to understand in a 3D model. As a result, a 3D model was built to explore geological conditions under the Earth's surface. With limited geological data because of cost controls, such as several boreholes, we were able to build the whole plane with spatial interpolation methods. It is a significant and easy way to show the distribution of the discrete points. However, it is mostly used for underground data analysis. The geological situation on the Earth's surface is often explored by geological engineers in the geological survey. It requires many computation resources because of the rendering in the 3D visualization model. It is a time-consuming method with low accuracy in some cases.

Image powered methods have become increasingly popular recently. These provide a novel method in geological structure identification. Vasuki et al. [8] captured rock surface images with unmanned aerial vehicles (UAVs) and detected rock features from the photos. According to the features detected on UAVs images, 3D models were built to show folds, fractures, cover rocks and other geological structures [9,10]. Furthermore, the automatic classification of geological images was also studied. The geological image, as a kind of unstructured data, contains much information including the object features. Młynarczuk et al. [11] applied four methods to make a classification of microscopic rock images automatically. The result of the nearest neighbor (NN) analysis showed high recognition level with 99.8%. Li et al. [12] also proposed a transfer learning method for sandstone identification automatically based on microscopic images. Shu et al. [13] used an unsupervised machine learning method to classify rock texture images. The experimental results indicated the outstanding performance of self-taught learning. Geological structures identification has many similarities with rock classification, which indicates what features we should extract from geological structures images. The color and texture are both critical in rock images to both micro and regular images. In some cases, the rock mineral was able to be classified just by color. While the geological structures data has unique characters. The texture is addressed more by the geomorphometric analysis [14].

Deep learning is prominent in image processing. It was proposed by Hinton [15] and was further developed recently [16]. Because of the positive performance, it was used to analyze unstructured data in many areas, such as image classification [17] and semantic analysis [18]. In medicine, deep learning is also popular [19,20]. However, Kim et al. [21] thought the input was massive in deep learning. Deep learning was also applied in remote-sensing image scene classification [22] and terrain features recognition [23]. It was able to extract features from unstructured data and make a classification with high accuracy. The convolutional neural network (CNN) is an essential method in deep learning. Scholars were able to build different CNN models by adding different kernels, pooling layers, and fully connected layers. Palafox et al. [24] used different CNN architectures in detecting landforms on Mars which proved convenient to extract features from images. Nogueira et al. [25] also used convolutional neural networks to extract features of images then built a support vector machine (SVM)

linear model based on the features. Xu et al. [26] also used transfer learning to predict geo-tagged field photos based on convolutional neural networks. All the results showed good performance of the convolutional neural network in extracting images features. While we should consider that the CNN model depends on a large dataset to avoid overfitting.

In this research, we established a transfer learning model based on Inception-v3 for geological structures with 12 labels. The test result showed a high accuracy of the model. Then we made a comparison between the identification model and the other four models, namely K nearest neighbors (KNN), artificial neural network (ANN), extreme gradient boosting (XGBoost) and CNN. The result showed the transfer learning model had a high accuracy on a small dataset. The machine learning method's accuracy was poor because it is hard to extract accurate features of images from a pixel vector or histogram. CNN was overfitting strongly. Transfer learning based on deep learning model was an effective method for geological structure images classification. Moreover, Wu et al. [27] applied the UAV and recognition model to detect rail surface defects. The retrained model in this research can also be combined with a UAV, which can be an assistant tool in the geological survey in further study.

## 2. Data Collection

### 2.1. Data Information

In this research, we collected 2206 geological structures images with 12 labels, including anticline, ripple marks, xenolith, scratch, ptygmatic folds, fault, concretion, mudcracks, gneissose, boudin, basalt columns, and dike. The dataset was collected from the geological survey and the internet [28]. In data collection, we tried to make each category cover images with different scales and sizes as many as possible. The resolution of the image is not limited. All the images are going to be processed at the same size before training. The numbers of images in each label are listed in Table 1. Figure 1 shows samples of the data.

**Table 1.** Information of geological structures images dataset.

| Geological Structure | No. | Geological Structure | No. | Geological Structure | No. |
|---|---|---|---|---|---|
| Anticline | 179 | Ptygmatic folds | 162 | Gneissose structure | 206 |
| Ripple marks | 221 | Fault | 127 | Boudin | 190 |
| Xenolith | 208 | Concretion | 181 | Basalt columns | 196 |
| Scratch | 164 | Mudcracks | 181 | Dike | 191 |

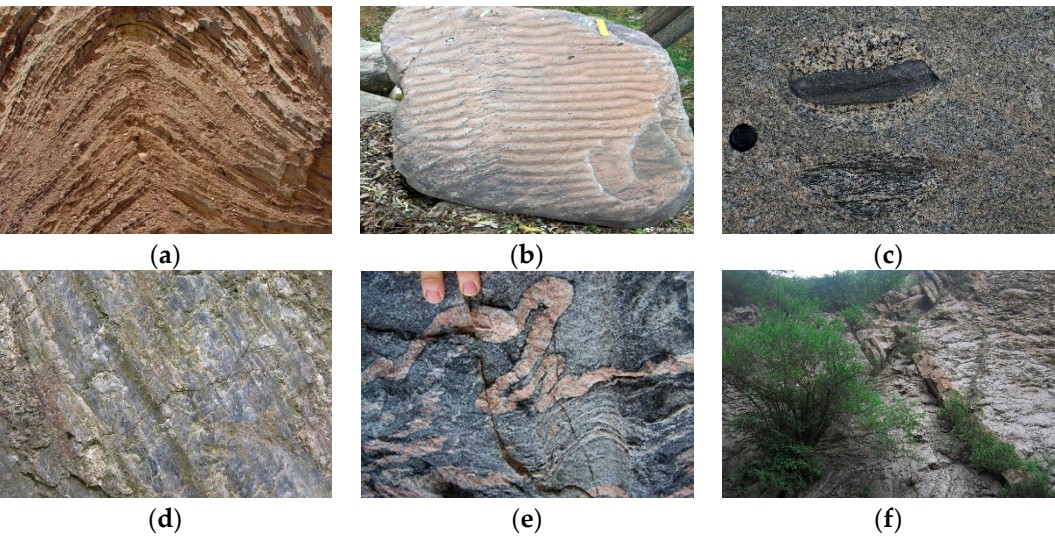

(a)     (b)     (c)

(d)     (e)     (f)

**Figure 1.** *Cont.*

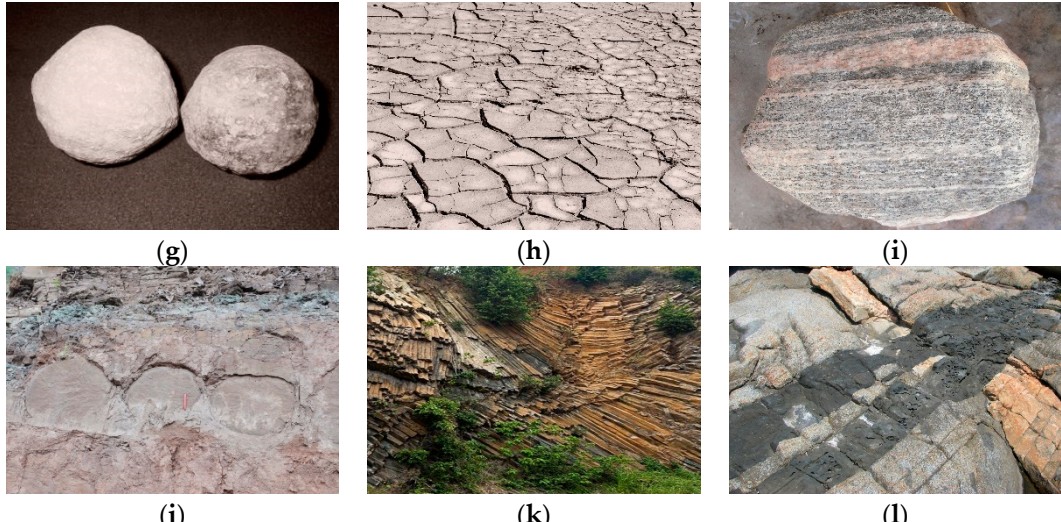

**Figure 1.** Samples of geological structure images: (**a**) Anticline; (**b**) Ripple marks; (**c**) Xenolith; (**d**) Scratch; (**e**) Ptygmatic folds; (**f**) Fault; (**g**) Concretion; (**h**) Mudcracks; (**i**) Gneissose structure; (**j**) Boudin; (**k**) Basalt columns; (**l**) Dike.

## 2.2. Data Preprocessing

It is necessary to make data preprocessing in images classification. Some feature pre-processing methods [29,30] were adopted to improve the performance of the model. We applied two direct and straightforward preprocessing methods to extract features in images as the input of KNN, ANN, and XGBoost. The first method is to convert pixels in each image into a row vector directly; the second method is to build the histogram of pixels based on their statistical characteristics, as shown in Figure 2. In Figure 2b,d, *x*-axis means the range of pixels, which is [0, 225]; *y*-axis means the numbers of pixels at each level. The color images have three channels, namely red, green, and blue; the grayscale images just have one channel-grayscale. In Figure 2b, the red, green, and blue lines refer to the numbers of the pixels at R, G, and B levels. In Figure 2d, The red line refers to the numbers of the pixels at the grayscale level. In Figure 2a,c, the *x*- and *y*-axis measure the size of the photo.

The features extracted from color, and grayscale images were both set as the input in KNN, ANN, and XGBoost, which was able to show the influence of image color. In CNN and transfer learning based on Inception-v3, the convolutional neural network was applied to extract the features of the images.

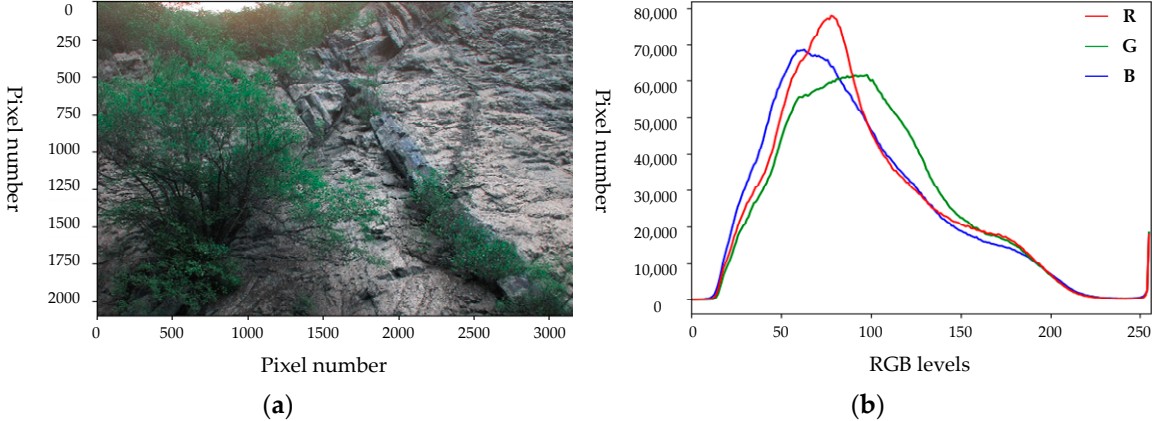

(**a**)　　　　　　　　　　　　　　　　　(**b**)

**Figure 2.** *Cont.*

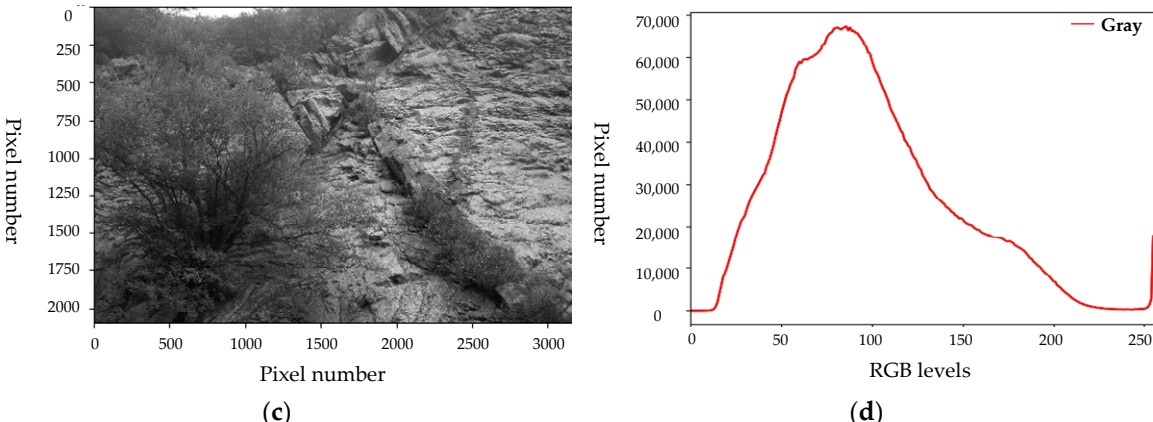

**Figure 2.** The histogram features extracted from color and grayscale geological structure image: (**a**) Color image; (**b**) Color image histogram; (**c**) Grayscale image; (**d**) Grayscale image histogram.

The raw data is not enough for a training model using CNN and transfer learning. As a consequence, some data augmentation methods, such as channel shift, shear, flip from left to right and flip from top to bottom, were adopted to raw data, as shown in Figure 3. A channel shift means to change the general color of the whole image; shear means to keep the horizontal axis (or vertical axis) stable and translate the other axis at a ratio. The translation distance is proportional to the distance to the horizontal axis.

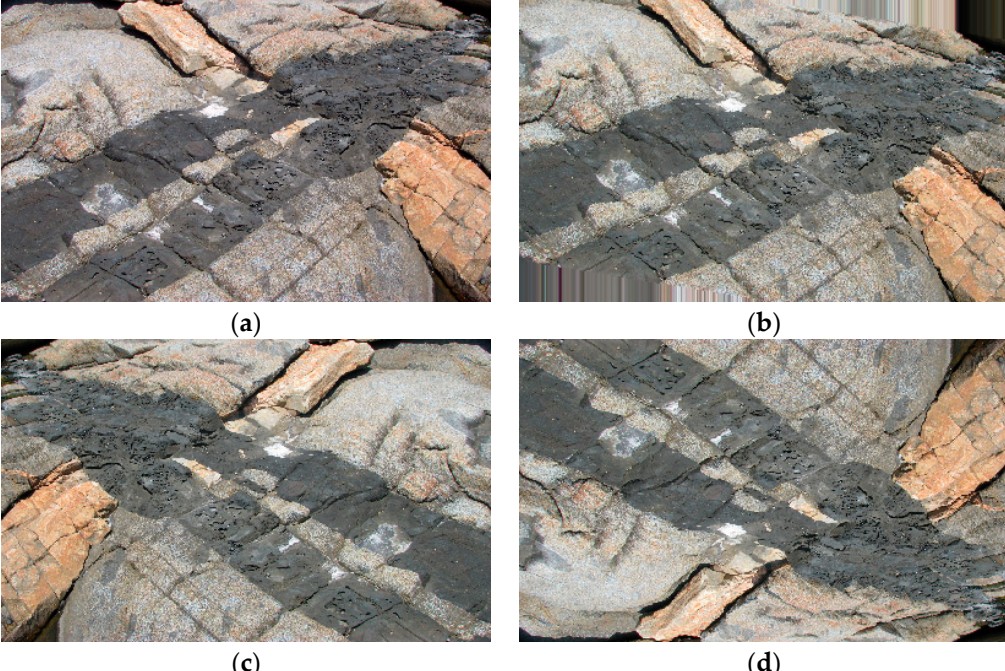

**Figure 3.** Data augmentation: (**a**) Channel shift; (**b**) Shear; (**c**) Flip from left to right; (**d**) Flip from top to bottom.

## 3. Key Techniques and Methods

### 3.1. Machine Learning

KNN is a lazy algorithm with no parameters, and ANN is a kind of supervised learning method. The two methods are used widely in prediction. XGBoost [31] is an improved gradient boosting decision tree (GBDT) method. In GBDT, the weak learners are combined to be a strong learner, as in the following equation:

$$f_i^{(t)} = \sum_{k=1}^{t} f_k(x_i) = f_i^{(t-1)} + f_t(x_i), \tag{1}$$

where $f_t(x_i)$ is the weak learner, namely a single decision tree; $f_t(x_i)$ is the sum of the weak learners. In each iteration, the new decision tree was added to the model. XGBoost improves the loss function, and also regularizes objective function, as shown in the following equation:

$$\begin{cases} L = \sum_{k=1}^{n} l(\bar{y}_i, y_i) + \sum_{k=1}^{t} \Omega(f_i) \\ \Omega(f) = \gamma T + \frac{1}{2}\lambda\|\omega\|^2 \end{cases}, \tag{2}$$

where $l$ is the loss function, which is used to measure the difference between prediction $\hat{y}_i$ and target $y_i$. $\Omega$ is used to control the complexity of the model. $\omega$ is the score of the leaves, $\lambda$ is the parameter for regularization, which is used to evaluate the node split.

### 3.2. Convolutional Neural Network (CNN)

CNN is a kind of feedforward neural network. The neurons in CNN can respond to the specific region in an image to extract features, which makes it outstanding in processing large unstructured data. A convolutional neural network includes convolutional layer, pooling layer, and fully connected layer.

There are three critical concepts in CNN, namely receptive field, parameter sharing, and pooling layers. Altenberger and Lenz [32] explained them in detail. The receptive field is a square region. It is a local subset of the neurons that the kernel connected to. The size of this square is the receptive field. Neurons of the same kernel should get the same pattern of the image regardless of their positions. As a result, the parameters should be shared by all the neurons of the same kernel. This concept is called parameter sharing. Pooling layers are also connected to a square region of the previous layer. However, polling layers are different from the convolutional layers. They are not determined by the weights or bias in the learning process, and the result just depends on the input. The max pooling is the common type in CNN. The maximum value that the neurons return is taken as the feature of the image. The average pooling can be interpreted in a similar way. Pooling reduces the complexity and dimensions of the feature map and improves the result to lead to less overfitting. At the same time, the features can keep translation invariance after pooling, which means if there are some translations, such as rotation, scale, distortion, in images, the pooling features are also effective.

As shown in Figure 4, the sizes of the image and convolutional layers are $5 \times 5$ and $3 \times 3$, respectively. There are nine parameters in the convolutional layer, namely the weight matrix. The nine parameters mean nine neurons. According to the sizes of image and kernel, the output is going to be a $3 \times 3$ matrix, which is called the feature map. In the first step, the neurons were connected to the receptive field on the image; then it slides to the next region by one stride in the second step, as shown in Figure 4b. The computation was processed in each neuro as follows.

$$f(x) = act\left(\sum_{i,j}^{n} \theta_{(n-i)(n-j)} x_{ij} + b\right), \tag{3}$$

where $f(x)$ was the output, *act* is the activation function, $\theta$ is the weight matrix, $x_{ij}$ is the input, $b$ is the bias. The softmax activation function is selected in this research.

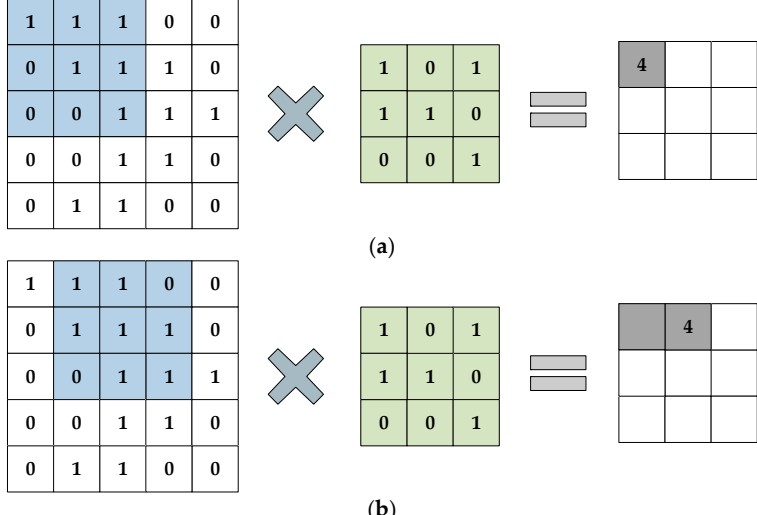

(a)

(b)

**Figure 4.** The process of convolutional neural network computation: (**a**) Computation in the first step; (**b**) Computation in the second step.

### 3.3. Transfer Learning

Even if there is an established model in a similar area, the model is going to be established from scratch using a machine learning method. It costs much manpower and time to solve problems individually in a similar domain. Considering the similarity between different tasks, we are able to build the model based on the knowledge obtained using transfer learning method. The knowledge obtained can be used again in a related domain with small change. If the gained knowledge can work in most cases or the data is hard to collect in the new task, we can make the most of the gained knowledge with transfer learning to build the new model. It benefits much in reducing dependency on big data and establishing a new model using a transfer learning method.

Furthermore, it is necessary to have a high-performance computer and time to train big data. However, it can reduce time cost and dependency on big data using transfer learning based on the pre-trained model [26,33,34]. We can apply a pre-trained model, which contains parameters trained by another big dataset, in training a new model in a similar domain. The kernels in the convolutional neural network can extract features of images automatically and effectively.

In this research, we adopted Inception-v3 [35] as the pre-trained model. The dataset which is used to train Inception-v3 contains 1.2 million images with more than 1000 labels. In the result of recognition, the top-5 accuracy in Inception-v3 is 96.5%, which is better than humans, with an accuracy of 94.9%. The convolutional layers and pooling layers in Inception-v3 can extract features from images as 2048 dimensional vectors. We removed the softmax layer in Inception-v3 and retained the new layer in our own domain. All the convolutional layers and pooling layers in Inception-v3 were used in extracting features from images, a process shown in Figure 5.

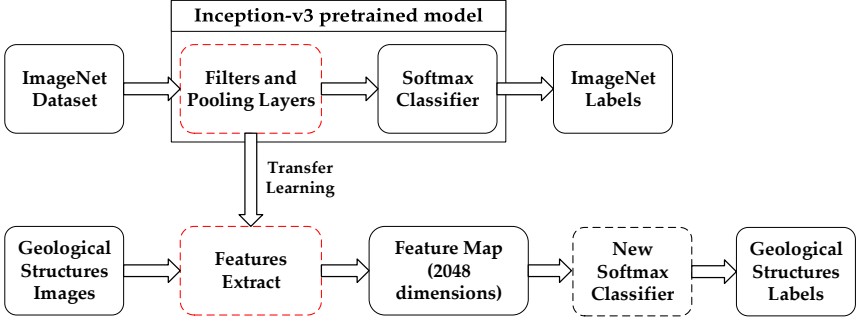

**Figure 5.** The process of retraining Inception-v3.

## 4. Model Establishment

### 4.1. Parameters Set

In the process of Inception-v3 retraining, iteration was set as 20,000; learning rate was set 0.01. In each iteration, 100 images were selected randomly to train the model, namely the training batch size equals 100. Batch size is limited by computer performance; 10% of the data was set as the test dataset. After 10 iterations, the model was evaluated. All the images were going to be used in the training process. Training accuracy, validation accuracy, and cross-entropy were used to evaluate the model in the training process. Training accuracy is gained by testing the trained dataset with the model; validation accuracy is gained by testing the validation dataset with the model; cross-entropy shows the model performance in identification. The smaller cross-entropy indicates a better performing model. In each training step, the prediction value and target value were measured to update the weight matrix. The geological structures images were cut as the same size before training. As a consequence, there were no strict limitations on the size and resolution of the images. The color images and grayscale images were both used in training, which was able to show the influence of color on the model.

In the training of CNN, we set two, three, and four convolutional layers and two fully connected layers to establish the model. The convolutional layers were set as $5 \times 5$ and $3 \times 3$, respectively. There were 64 neurons in each convolutional layer. While there were 128 neurons in fully connected layers. The learning rate was set as $10^{-4}$. Batch size was set as 32. The data was split into training data and validation data; 80% of the data was set as training data; 10% data was set as validation data and test data.

In the model establishment of KNN, ANN, and XGBoost, we used OpenCV [36] to process the raw data into two datasets, namely color images and grayscale images, with the size of $128 \times 128$. The two datasets were used to build images features with origin pixels and pixels histogram. Then we took the pixel vectors and histogram features as the input of KNN, ANN, and XGBoost. The python package Scikit-learn [37] was used in the research to build the three models, and all the parameters of the models were set as in Table 2.

**Table 2.** Parameters in K nearest neighbors (KNN), artificial neural network (ANN), and extreme gradient boosting (XGBoost).

| Method | Parameters | Value |
|--------|------------|-------|
| KNN | n_neighbors | 1 |
| | p | 2 |
| XGBoost | colsample_bytree | 0.8 |
| | learning_rate | 0.1 |
| | eval_metric | mlogloss |
| | max_depth | 5 |
| | min_child_weight | 1 |
| | nthread | 4 |
| | seed | 407 |
| | subsample | 0.6 |
| | objective | multi:softprob |
| ANN | hidden_layer_sizes | 50 |
| | max_iter | 1000 |
| | alpha | $10^{-4}$ |
| | solver | sgd |
| | tol | $10^{-4}$ |
| | random_state | 1 |
| | learning_rate_init | 0.1 |

### 4.2. Model Train and Test

Figure 6 showed the training process of the transfer learning. The model was evaluated by train accuracy, validation accuracy, and cross-entropy. In Figure 6, train accuracy, and validation accuracy both increased gradually. Then train accuracy converged to about 97.0% and validation accuracy converged to about 90.0%. Cross-validation decreased gradually and converged to about 0.2. Finally, the test accuracy based on grayscale and color images were 91.0% and 92.6%, respectively. The small difference between the two models indicates that color had little influence on the model identification for geological structures, which means textures are more important in identification.

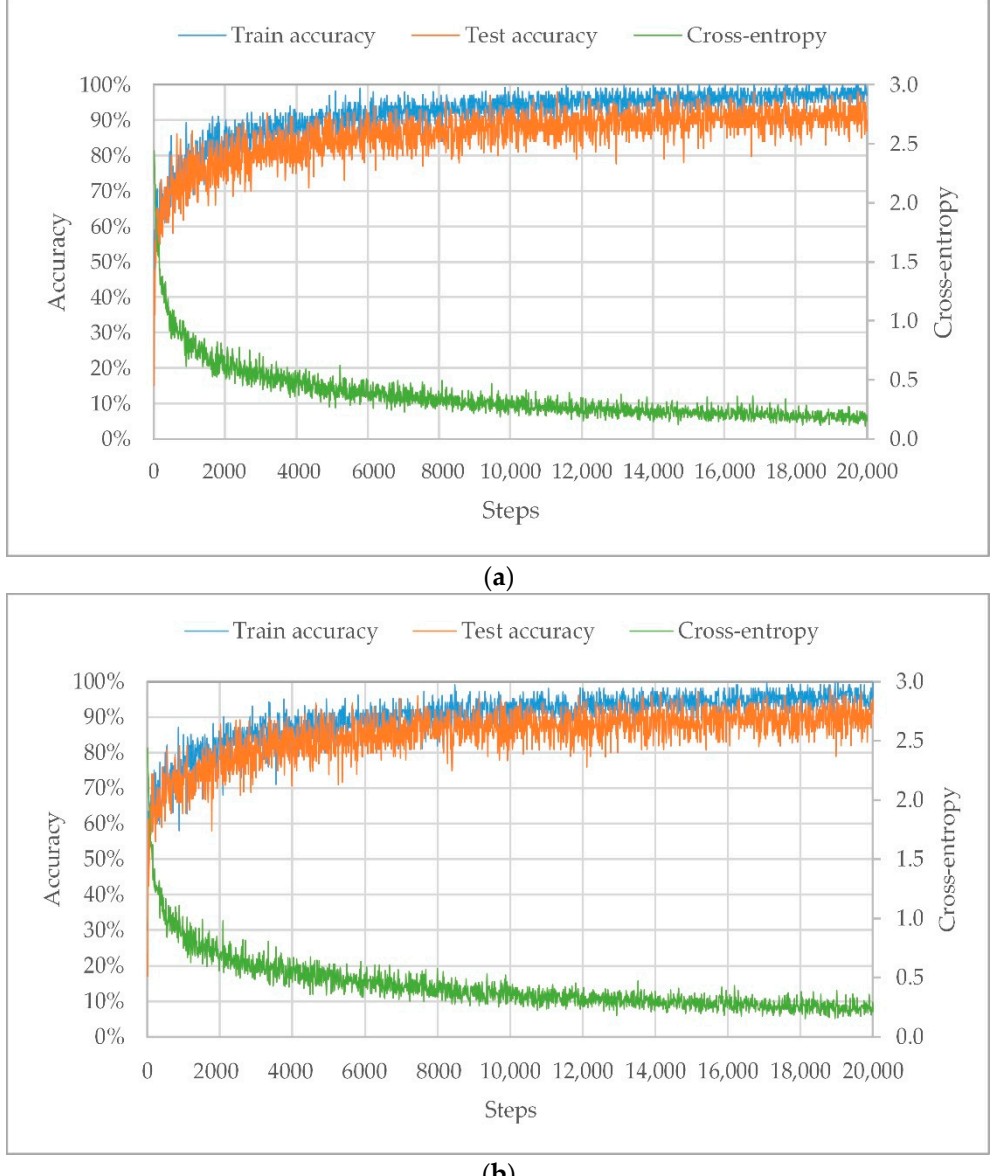

**Figure 6.** Train accuracy, validation accuracy and cross-entropy variation in transfer learning process: (**a**) Grayscale dataset; (**b**) Color dataset.

The patches from the same image are similar using data augmentation. At the same time, we want to apply the model to identify geological structure images from the geological survey. So we chose another 60 images which were from an engineering project to test the model accuracy. Top-1 and top-3 accuracy were used in model evaluation. Top-1 accuracy means the prediction with the



highest probability matches the right label. Top-3 accuracy means any one of predictions with the three highest probability matches the right label. In the model testing, the top-1 accuracy was 83.3% and the top-3 accuracy was 90.0%. The test images in Figure 7 are the same as those in Figure 1. In Figure 7, the result showed top-3 prediction probability. In the identification of faults, the top-1 and top-3 result were wrong. The number of fault images should be increased. In the identification of boudin images, the top-1 prediction was wrong. However, we found that the probability of boudin was 15.7%, which ranked third. As a consequence, it is better to apply top-1 and top-3 accuracy to evaluate the result comprehensively in predictions.

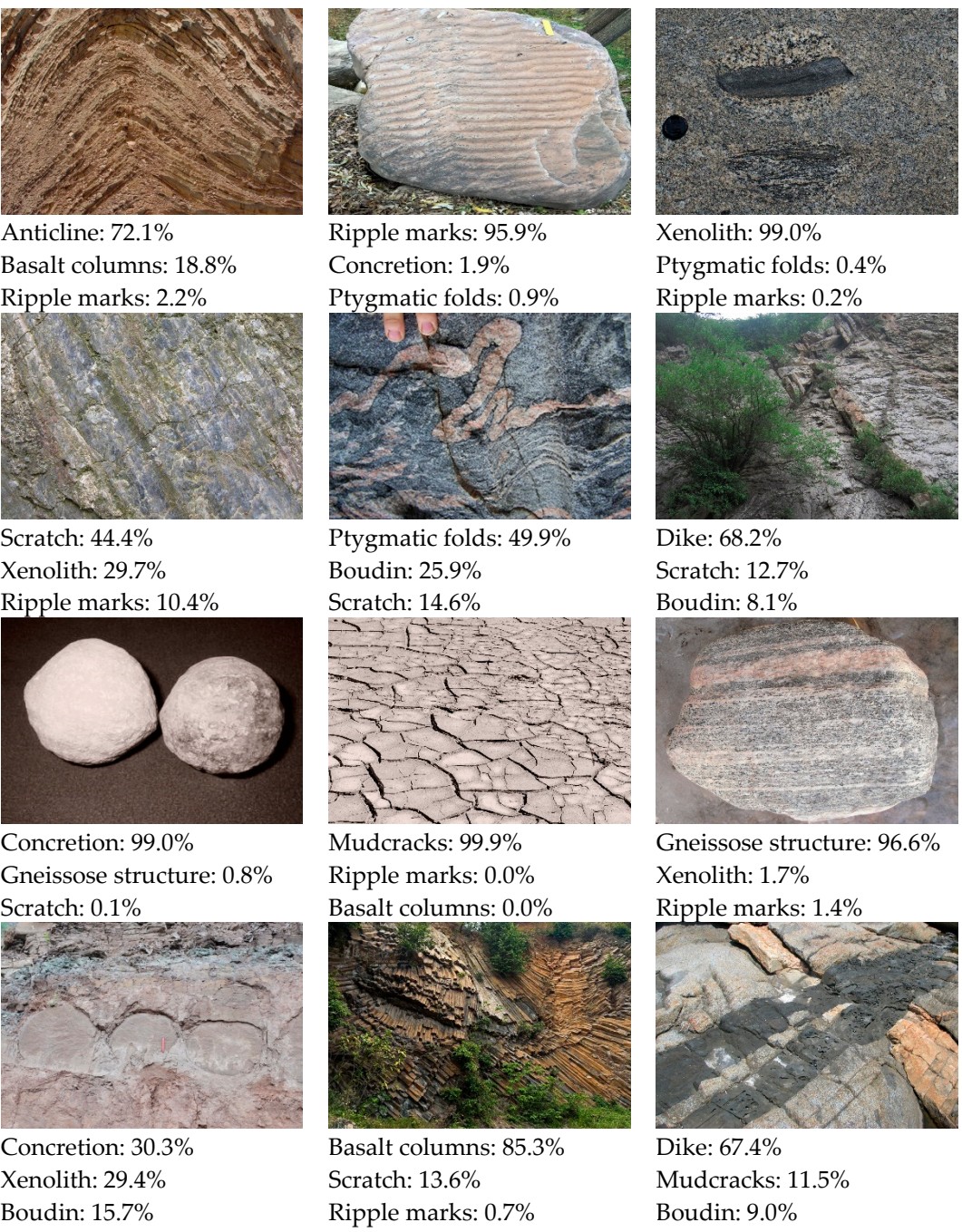

**Figure 7.** Identification of geological structures images results.

The training process of CNN was shown in Figure 8. Train accuracy, validation accuracy, and cross-entropy were also used to evaluate the model. Figure 8a–c are the results of the CNN with two, three and four convolutional layers on the color dataset; Figure 8d is the result of the CNN with three convolutional layers on the grayscale dataset. The effects of the three-layer CNN was the best, and the grayscale data was also trained by the CNN architecture, as shown in Figure 8d. The train accuracy was almost 100.0%, while the validation accuracy was about 85.0%, which indicated the model was overfitting.

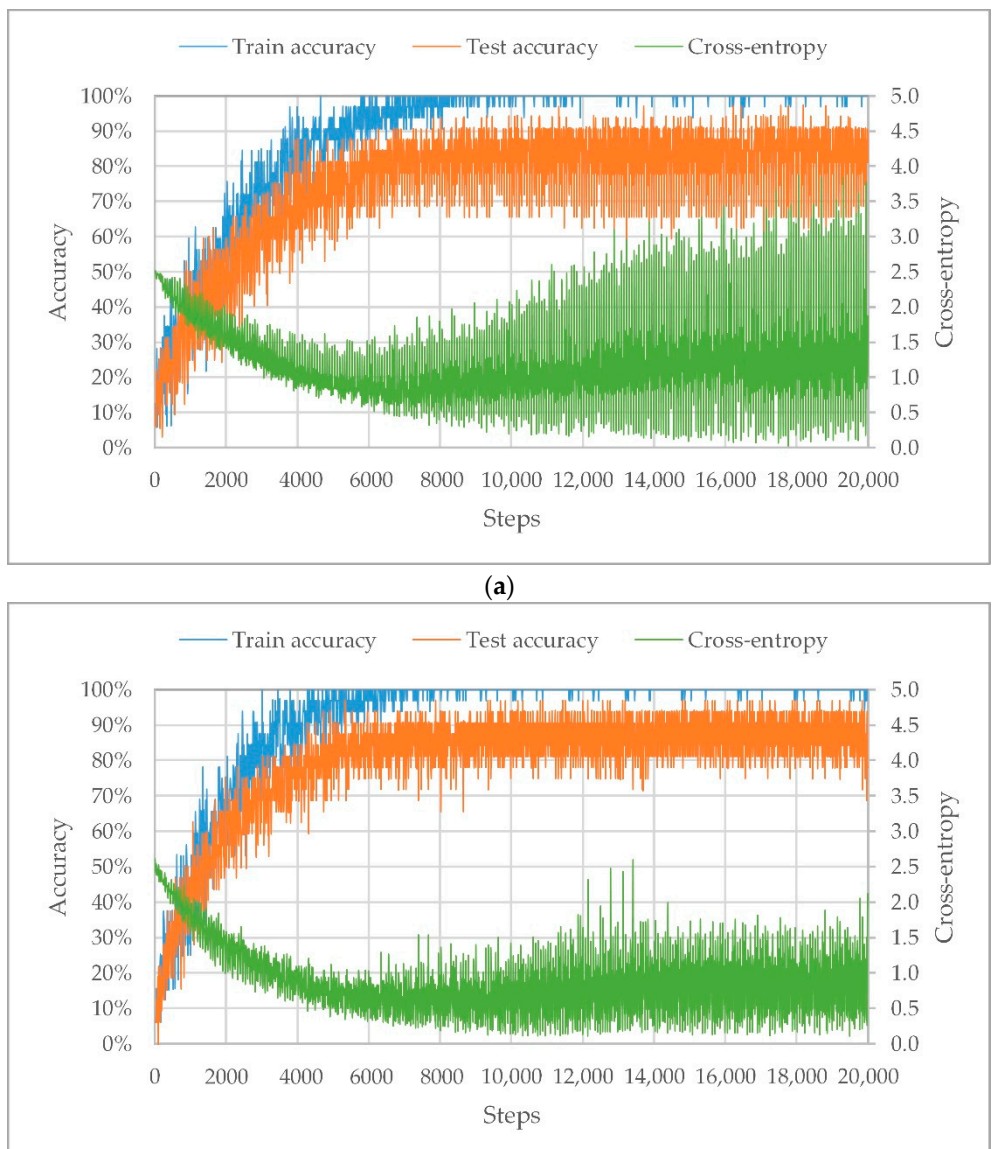

**Figure 8.** *Cont.*

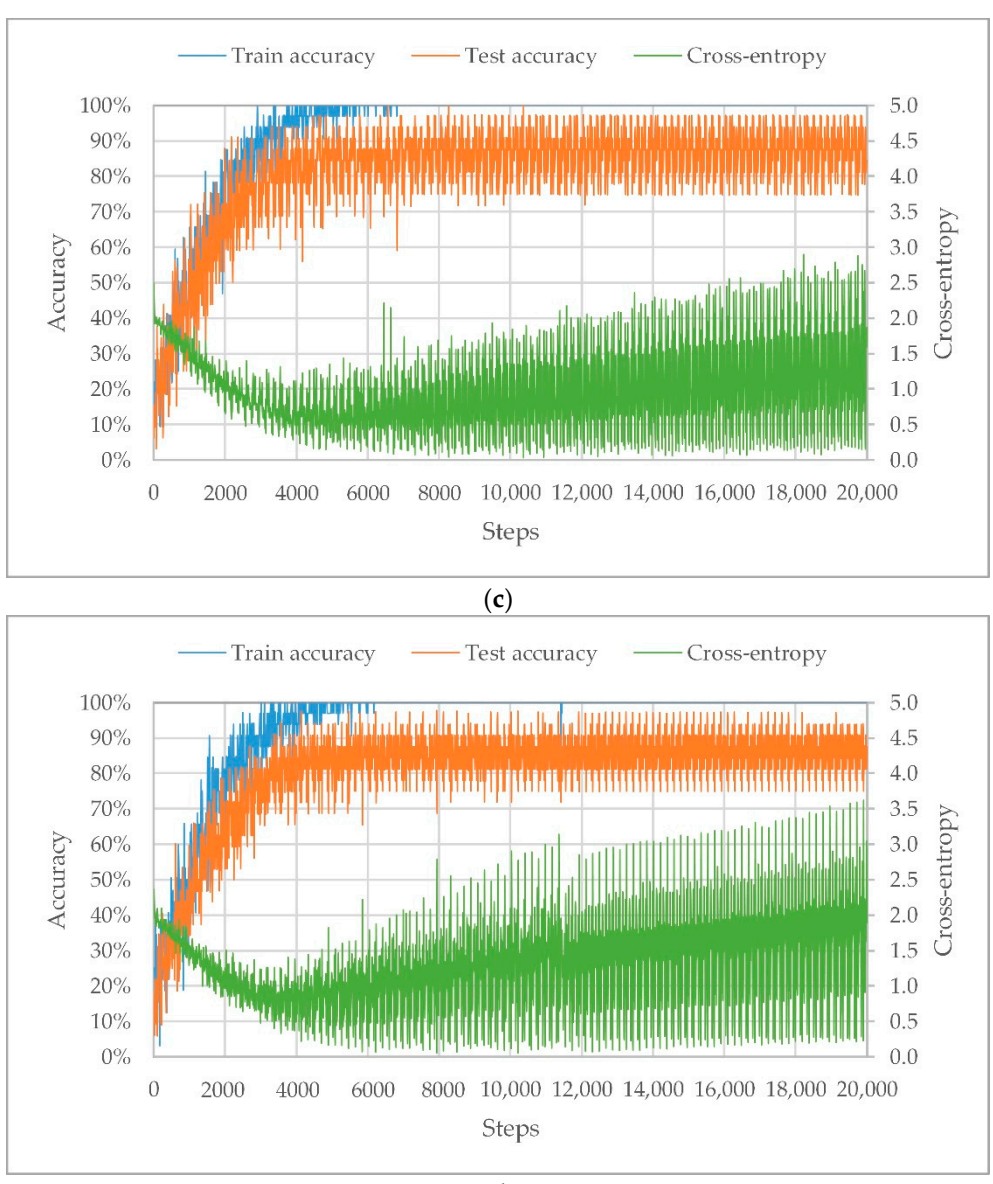

**Figure 8.** Train accuracy, validation accuracy and cross-entropy variation in the convolutional neural network (CNN) training process: (**a**) Two layers using color dataset; (**b**) Three layers using color dataset; (**c**) Four layers using color dataset; (**d**) Three layers using grayscale dataset.

The results of the five methods were summarized in Table 3. The accuracies of KNN, ANN, and XGBoost models were less than 40%, while the KNN and XGBoost also showed better performance in color images with histogram features. The test accuracy of CNN model based on grayscale and color images was 80.1% and 83.3%, but the model was overfitting; the test accuracy of the transfer learning model based on Inception-v3 model reached 91.0% and 92.6%, which was the best in all the methods. The result indicated the convolutional layers and pooling layers in Inception-v3 were able to extract features from geological structures images effectively. As a consequence, the transfer learning method was chosen to identify the geological structure image from engineering. The top-1 and top-3 accuracy were 83.3% and 90.0%, respectively.

**Table 3.** Comparison result between the five methods.

| | Grayscale Image Feature | | Color Image Feature | |
|---|---|---|---|---|
| | **Pixel** | **Histogram** | **Pixel** | **Histogram** |
| KNN | 20.4% | 19.6% | 20.4% | 33.4% |
| ANN | 9.1% | 19.3% | 9.4% | 31.4% |
| XGBoost | 25.2% | 20.7% | 33.4% | 34.8% |
| Three-layer CNN | 80.1% | | 83.3% | |
| Transfer Learning | 91.0% | | 92.6% | |

### 4.3. Discussion

In this research, we built a geological structure identification model based on Inception-v3. In a comparison between KNN, ANN, XGBoost, and CNN, the convolutional layers and pooling layers in Inception-v3 were effective in extracting features from images of the small dataset. Actually, the small geological structures dataset we used in the research has its own characters. For example, the boudins in Figure 9a–c are very different, even though they have the same label. On the other hand, boudin and xenolith are with different labels; however, they are similar and easy to be mixed in some cases, as shown in Figure 9c,d. In Figure 7, the identification result of boudin also proved that. Meanwhile, the prediction shows the probability of xenolith is 29.4%, which is higher than that of boudin. The result shows that the features of boudin and xenolith are similar in some cases.

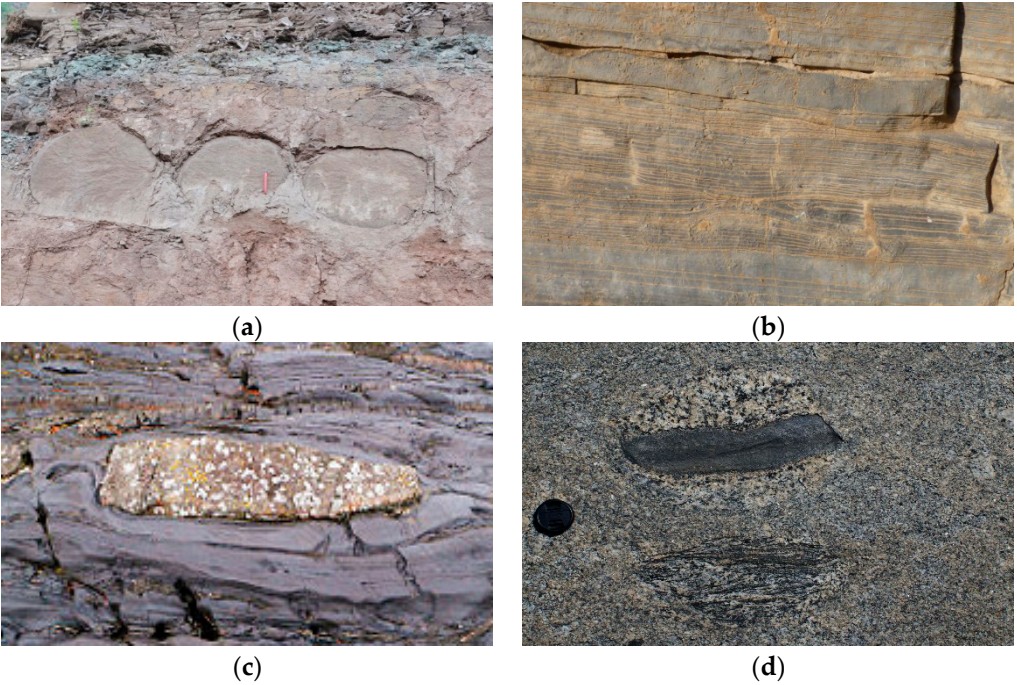

**Figure 9.** (**a**–**c**) Boudin; (**d**) Xenolith.

We also built a CNN model to establish identification. Because the dataset was small, we designed a simple net with two convolutional layers and two fully connected layers, while the model was still overfitting. The single architecture of CNN did not work in any cases. In ConvNets, Palafox et al. [24] also designed several CNN models with different architectures in different cases. The retrained model based on Inception-v3 can extract image features effectively with the convolutional layers and pooling layers. It is not necessary to redesign model architecture in the transfer learning model, and it worked well on the small dataset in this research. It was hard to extract real image features based on pixel vector and histogram in which translation invariance could not be kept. The background noise also interfered the features extraction significantly. As a result, the inaccurate input led to a low accuracy

in KNN, ANN, XGBoost. Actually, some feature pre-processing methods can be applied in model training. The restricted Boltzmann machines (RBM) pre-processing method [29] and the mixture of handcrafted and learned features [30] can improve the model performance. Model ensemble [38] is also a robust method to enhance the feature extraction from images. Model ensemble and feature engineering are going to be applied in the model establishment in the further study.

## 5. Conclusions

In this research, we built the geological structure identification model based on a small dataset and the Inception-v3 model. The grayscale and color images datasets were both trained to construct different models. The two models had an accuracy of 91.0% and 92.6%, respectively. At the same time, we used 60 engineering images to test the model. The top-1 and top-3 accuracies were 83.3% and 90.0%, which showed the kernels and pooling layers in the Inception-v3 model could extract image features effectively. CNN models with different layers were built as well, while the model was overfitting in training even just with two convolutional layers and two fully connected layers. Three convolutional layers were adopted to establish the model in our study. The best parameters in CNN are hard to reach because it depends on experience. We also used OpenCV to build pixel feature based on origin pixel information and a pixel's histogram. However, the images features could not be extracted accurately in this way, which led to low accuracy in KNN, ANN, and XGBoost models. More feature engineering methods should be considered in the future. The retrained models based on Inception-v3 were trained using transfer learning method with color and grayscale datasets and had a small difference in accuracy, which indicated that color had little influence on geological structure identification.

There are also some weaknesses in the model trained by a small dataset. Test data is small and overfitting still exists in the training process. Even though data augmentation was adopted, some patterns and features were not learned by the model. In this research, we proved the feasibility of transfer learning for geological structures classification. If the model is applied in practice in the future, more data should be added.

Transfer learning based on the Inception-v3 model has strong adaptability for a small dataset. In the future, we are going to extend our dataset and combine the model with a UAV, which can be applied as a tool in geological surveys.

**Author Contributions:** Y.Z. wrote the code and the paper; G.W. gave professional geological guidance and the image data; M.L. provided the idea and edited the manuscript; S.H. collected and analysed the data.

**Funding:** This research was funded by the Tianjin Science Foundation for Distinguished Young Scientists of China, Grant no. 17JCJQJC44000 and the National Natural Science Foundation for Excellent Young Scientists of China, Grant no. 51622904.

**Conflicts of Interest:** The authors declare no conflict of interest.

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
