# Peer review of "Automated Classification Analysis of Geological Structures Based on Images Data and Deep Learning Model"

_applsci, doi:10.3390/app8122493_

Reviewer 1 Report

In this paper, the authors used ~2K labeled images to make geological structures identification based on Inception-v3 model. KNN, ANN, and XGBoost were applied in geological structures classification based on extracted features, and a CNN was developed. Based on their results, the authors claim that KNN, ANN, and XGBoost had a poor performance. 

This is well-written paper and the concept is well presented. However, there are two main concerns that raised when reading this paper. Firstly, it is quite important to make a case why the classification process of such images is important, and what are the benefits when correctly classifying such images. Secondly, how your CNN outperform the current CNN architectures (or developed methods)?

Regarding the first concern, I would suggest authors consider a wide range of applications that the classification of geospatial structures. Thus, less experienced readers would be able to understand even more the applicability and the domain. You can consider including examples like the "Environment-Scale Fabrication: Replicating Outdoor Climbing Experiences" in which rock climbing walls are reconstructed. For example, classifying the geospatial properties rock climbing walls can help us find climbing walls with different difficulties.

For the second concern, I would suggest authors evaluate their method with different CNN architecture to understand how their method actually performs.

I would also like to ask the authors if they considered pre-processing the extracted features before using ML to classify them. It is well known that when pre-processing the extracted features the accuracy of an ML technique can be increased. Take a look at the paper entitled "Learning Motion Features for Example-Based Finger Motion Estimation for Virtual Characters" which preprocess the motion features using RBMs. Please consider adding and discussing the feature pre-processing at the revised manuscript.

I feel confident that after a major revision the paper should be ready for publication

Reviewer 2 Report

The proposed method manages an interesting problem, my suggestions are:

·         If there are not copyright problems please make the tested dataset available

·         Update the introduction on CNN using https://arxiv.org/pdf/1803.02129.pdf

·         You have compared CNN with very low performance handcrafted approaches, in the literature several ensemble of high performing texture descriptors are proposed , e.g. in https://www.sciencedirect.com/science/article/pii/S0031320317302224 is reported a set of handcrafted methods that work comparably to CNN in small/medium size training set

·         You should better compare your transfer learning approach with other recent texture classiifcation methods (e.g. https://www.researchgate.net/publication/325786577_Ensemble_of_Convolutional_Neural_Networks_for_Bioimage_Classification  )

·         Do you have performed a single split training/test? It is a not reliable way to assess the performance, usually a n-fold cross validation (n=5 or n=10) is used  https://en.wikipedia.org/wiki/Cross-validation_(statistics)#k-fold_cross-validation

Reviewer 3 Report

The paper presents an application of CNN for image classification of geological structures.
The following are some suggestions to improve the quality of the paper prior publication:
1. Authors mentioned that the images are also collected from internet. Did the Authors cite the weblink of the source images? If no, I suggest to put it in the References with information when it was accessed.
2. What are the x- and y-axis in Figure 2(b) and (d)? please also provide a unit.
3. Please also provide a legend of Figure 2(b). What the red, green and blue line refer to?
4. A bit difficult to understand Table 2. The subsample: 0.6 and objective multi:softprob are belong to which method? it is KNN or XGBoost?
5. Poor resolution in Figure 6. Please reproduce Figure 6 with better resolution.
6. Similar to comment No.4. Please reproduce Figure 8.

Author Response

Round  2

Reviewer 1 Report

The requested revision has been made by the author, therefore I am recommending this paper for publication at the Applied Sciences Journal.

Reviewer 2 Report

Accept in present form

Reviewer 3 Report

Dear Authors,

Thank you for providing a revised manuscript.

Kind regards,

- Reviewer -